# Neurobiology of the Orexin System and Its Potential Role in the Regulation of Hedonic Tone

**DOI:** 10.3390/brainsci12020150

**Published:** 2022-01-24

**Authors:** Martin A. Katzman, Matthew P. Katzman

**Affiliations:** 1Stress, Trauma, Anxiety, Rehabilitation and Treatment (START) Clinic for the Mood and Anxiety Disorders, Toronto, ON M4W 2N4, Canada; katzmanmatthew@gmail.com; 2Department of Psychology, Adler Graduate Professional School, Toronto, ON M4W 3P4, Canada; 3Department of Psychiatry, Northern Ontario School of Medicine, Thunder Bay, ON P7B 5E1, Canada; 4Department of Psychology, Lakehead University, Thunder Bay, ON P7B 5E1, Canada; 5Department of Psychology, Queens University, Kingston, ON K7L 3N6, Canada

**Keywords:** orexin, sleep, arousal, reward, motivation, hedonic tone, depression, anxiety, ADHD, anhedonia

## Abstract

Orexin peptides comprise two neuropeptides, orexin A and orexin B, that bind two G-protein coupled receptors (GPCRs), orexin receptor 1 (OXR1) and orexin receptor 2 (OXR2). Although cell bodies that produce orexin peptides are localized in a small area comprising the lateral hypothalamus and adjacent regions, orexin-containing fibres project throughout the neuraxis. Although orexins were initially described as peptides that regulate feeding behaviour, research has shown that orexins are involved in diverse functions that range from the modulation of autonomic functions to higher cognitive functions, including reward-seeking, behaviour, attention, cognition, and mood. Furthermore, disruption in orexin signalling has been shown in mood disorders that are associated with low hedonic tone or anhedonia, including depression, anxiety, attention deficit hyperactivity disorder, and addiction. Notably, projections of orexin neurons overlap circuits involved in the modulation of hedonic tone. Evidence shows that orexins may potentiate hedonic behaviours by increasing the feeling of pleasure or reward to various signalling, whereas dysregulation of orexin signalling may underlie low hedonic tone or anhedonia. Further, orexin appears to play a key role in regulating behaviours in motivationally charged situations, such as food-seeking during hunger, or drug-seeking during withdrawal. Therefore, it would be expected that dysregulation of orexin expression or signalling is associated with changes in hedonic tone. Further studies investigating this association are warranted.

## 1. Introduction

Neuropeptides orexin A and orexin B were first described in the late 1990s and were initially shown to regulate feeding behaviour [1,2]. Since their discovery, it has been shown that orexins are involved in diverse functions that range from the modulation of autonomic functions (i.e., food intake, sleep regulation, and cardiovascular function), and promotion of physical activity, as well as reward-seeking behaviour and higher cognitive functions, including attention, cognition, and mood [3,4]. While the roles of orexin in the regulation of food intake, sleep/wake cycles, and addictive behaviours have been extensively studied and reviewed [4,5,6,7,8,9,10], the role of these peptides in the modulation of mood and affect has not been as thoroughly explored. The aim of this review is to provide a general overview of the neurobiology and physiological roles of orexin, with a focus on its role in sleep as well as its activity as a mediator of reward pathways, mood, and affect. Additionally, this review will specifically discuss the potential overlap of the orexin projection system its effect on the modulation of hedonic tone.

## 2. Neurobiology of the Orexin System

Orexins comprise two neuropeptides, orexin A and orexin B, which are proteolytically cleaved from a precursor peptide, prepro-orexin [11]. Orexin A is a 33-amino acid peptide with an N-terminal proglutanyl residue, two intrachain disulfide bonds, and C-terminal amidation [11]. Orexin B is a 28-amino acid, C-terminally amidated linear peptide. The two peptides have very similar C-terminals and different N-terminals. Orexins were initially identified as ligands for orphan G-protein coupled receptors (GPCRs), which were later identified as orexin receptor 1 (OXR1) and orexin receptor 2 (OXR2) [1,2]. Orexin A has an equal affinity to both receptors, whereas orexin B has a 10-fold higher affinity for OXR2 than for OXR1 [12].

Although cell bodies containing prepro-orexin mRNA and orexin peptides are localized in a small area that includes the lateral hypothalamus and adjacent regions, orexin-containing fibres project throughout the neuroaxis, from the spinal cord to cortical regions [1,2]. Accordingly, OXR1 and OXR2 receptors are widely distributed throughout the central nervous system [13,14,15,16]. Although the distribution patterns of the two receptors overlap to a great extent, some regional differences have also been reported. For example, OXR1 is more highly expressed in cortical regions, the bed nucleus of the stria terminalis, and the locus coeruleus, whereas OXR2 expression is higher in the NAc [16].

## 3. Overview of Physiological Roles of the Orexin System

### 3.1. Role of Orexin in the Regulation of Sleep and Arousal

The role of orexin in promoting arousal and waking from sleep has been well documented, with evidence showing that orexin plays a critical role in the regulation of sleep-to-wake transitions [3,17]. Additionally, an orexin deficiency is observed in narcolepsy, a disorder characterized by excessive sleepiness, sleep paralysis, and in some cases, episodes of cataplexy [3,17].

Further to this point, orexin neurons have been shown to be more active during wakefulness than during sleep, and their stimulation causes waking within several seconds [18]. Physiological studies have shown that orexin lengthens, and orexin disruption shortens continuous periods of wakefulness. Furthermore, loss of orexin neurons or receptors are hallmarks of narcolepsy/cataplexy in animals and humans. Additionally, orexin projections have been shown to excite wake-promoting regions, including the locus coeruleus norepinephrine secreting neurons, dorsal raphe serotonin secreting neurons, tuberomammillary histamine secreting neurons, and basal forebrain/brainstem acetylcholine secreting neurons [18].

Neuroanatomical studies have further supported the role of the orexin system in sleep and wakefulness by demonstrating that orexin-containing neurons project to key brain regions involved in the regulation of sleep-wake cycles, including histaminergic neurons in the tuberomammillary nucleus (TMN), cholinergic and GABAergic neurons of the basal forebrain, dopaminergic neurons within the ventral tegmental area (VTA), and norepinephrine neurons in the LC [17]. In addition, arousal has been shown to occur from the direct stimulation of the mesopontine cholinergic nuclei and the locus coeruleus (LC), which contain noradrenergic neurons [19]. Additionally, neurons in the basal forebrain, an attention- and arousal-sustaining area, have been shown to be depolarized by orexin [20]. The basal forebrain receives projections from midbrain dopamine neurons, and it has been proposed that this connection may underlie the coupling of motivation to arousal states. Notably and in support, the VTA dopamine neurons, which have been implicated in motivation, have also been shown to be active in the promotion of arousal and the initiation of sleep-preparatory behaviours, also express orexin receptors. Additionally, administration of orexin A into the LC increases firing rate, whereas optogenetic inhibition of LC neurons blocks orexin-induced sleep-to-wake transition [21,22,23,24].

### 3.2. Role of Orexin in the Regulation of Affect, Motivation, and Reward

The role of the orexin system in the regulation of motivation and reward-seeking behaviours has been well established [3]. Evidence for the role of orexins in modulation of these function primarily comes from studies of food-seeking and drug-seeking behaviours and has been extensively reviewed by other authors [4,25,26]. Generally, this evidence indicates that the orexin system receives information about the internal environment, such as macronutrient or circulating hormone levels, blood pressure, and circadian rhythms, and integrates this information to coordinate behaviours and arousal levels that are appropriate for reward-directed behaviours, such as acquisition of food [4]. Additionally, the orexin system specifically mediates food seeking and consumption in motivationally charged circumstances, such as hunger or during Pavlovian cues [18].

The actions of orexins on motivation and reward-seeking are mediated by projections to areas of the mesolimbic reward pathway, including the VTA and the nucleus accumbens (NAc) [4]. Within the VTA, 20% of neurons that originate in the lateral hypothalamus display orexin immunoreactivity [27]. In the VTA, orexin-containing axons make appositional contacts onto both dopaminergic and GABAergic neurons. Further, orexin is co-localized with dopamine in neuronal fibres in the medial prefrontal cortex (mPFC) and the medial shell of the NAc, indicating that orexin may interact with dopamine in these reward areas [28]. Further, orexin neurons also project to the amygdala, a region that connects the basal forebrain to the classical reward systems of the LH (Schmitt et al., 2012).

### 3.3. Influence of Orexin on Attention/Cognition and Learning

Given that arousal and attention are closely linked, it is not surprising that orexin has also been implicated in the regulation of attention. Several lines of evidence indicate that orexin contributes to attentional processing and learning via actions on the mPFC, cholinergic system in the basal forebrain, dopaminergic neurons in the ventral midbrain, and noradrenergic neurons in the LC [19]. Studies have shown that administration of orexin into the basal forebrain excites cholinergic neurons, induces Ach release, and increases attention. Further, administration of orexins into the basal forebrain excites cholinergic neurons that release Ach in the cerebral cortex and thereby promotes wakefulness. Orexin has also been shown to improve attentional processes via actions in the medial prefrontal cortex (mPFC) [29]. In one study, administration of orexin B into the PFC improved accuracy under high attentional demand by exciting the same thalamocortical synapses that are activated by Ach from the basal forebrain [29]. These findings suggest that orexin may promote attention through an increased Ach release and direct actions on thalamo-cortical projections.

Orexins have also been shown to play a role in appetitive and motivated learning via actions in the mPFC, LC, and the hippocampus. In the mPFC, where they increase sustained attention and modulation of basal forebrain neurons [18]. In the LC, orexins have been shown to play a role in the acquisition of Pavlovian fear conditioning, and hippocampal orexins play a role in learning performance in a Morris water maze task. Further, orexin neurons are required for attentional aspects of motivated learning and acquisition of morphine-conditioned place preference requires orexinergic projections to the VTA. Finally, antagonism of orexin receptors in the VTA reduced cocaine-evoked premature responses in a five-choice serial reaction time task, indicating that orexin neurons are involved in executive function in normal and pathological conditions [30].

### 3.4. Influence of Orexin on Mood and Mood and Psychiatric Disorders

Given the roles of orexin in reward-seeking, arousal, and motivation, it has been hypothesized that dysregulation of orexin signalling has been demonstrated in psychiatric disorders that are characterized by deficits in these states. A study in humans showed that orexin-A levels are maximal during positive emotion, social interaction, anger, and increase at wake onset, whereas lowest orexin-A levels are observed during sleep, during wakefulness just before sleep, and during wakefulness while in pain, indicating that orexin-A levels are linked to specific emotions and state transitions [31]. Further, orexin signalling may be affected in ADHD, anxiety, depression, and addiction.

#### 3.4.1. Depression

Studies of the role of orexin in depression have reported that depression is associated with changes in the level of orexin and their receptors, although both increases and decreases in orexin levels have been reported (Deats et al., 2014; Nocjar et al., 2012; Nollet et al., 2011). One study using a rat model of depression showed that depression was associated with an increase in the number of orexin-positive neurons in the hypothalamus. However, a study of an animal model of seasonal affective disorder (SAD) reported a reduction in the number of orexin-A immunoreactive neurons in the hypothalamus and attenuated orexin-A fibre density in the dorsal raphe nucleus of animals with SAD compared with controls [32]. Further, treatment of animals with SB-334867, a selective OX1R antagonist, led to a depressive phenotype in control animals, indicating that loss of OX1R activation led to depressive symptoms [32]. Rats with depressive-like symptoms were also shown to have lower levels of orexin in the hypothalamus, mPFC, and VTA compared with controls, indicating that a reduction in orexin levels was associated with depressive-like symptoms [33]. In another study, animals with depressive-like symptoms had a lower expression of OXR2 in the thalamus and the hypothalamus compared with controls [34]. However, the same group reported that treatment with dual orexin receptor antagonist almorexant prevented the development of depressive-like symptoms, indicating that pharmacological blockade of the orexinergic system induces an antidepressant-like effect.

Studies in humans have also investigated changes in orexin levels in depressive disorders and after suicide attempts. In a study of 101 patients with a recent suicide attempt, low orexin-A level in the CSF was correlated with difficulties initiating activities (lassitude), a greater decrease in frequency and extent of voluntary movement, and a greater rating of global illness [35]. Another study by the same group showed that CSF levels of orexin-A were significantly lower in patients with MDD than in those with adjustment disorder and dysthymia after a suicide attempt [36]. Further, a follow-up study of ten patients showed that orexin-A levels were higher one year after compared to immediately after the attempt [37]. The authors concluded that suicidal patients with MDD may have reduced levels of orexin-A, a peptide regulating the state of arousal.

Another study that analysed 120 postmortem samples from patients with depression, some of whom died by suicide, reported that orexin A immunoreactivity was significantly increased in female, but not male patients with depression compared to controls [38]. Additionally, unlike controls, patients with depression did not exhibit the diurnal fluctuation in orexin immunoreactivity. Notably, male patients who died by suicide had significantly higher expression or OXR2 mRNA in the anterior cingulate cortex compared to male controls. Taken together, these findings indicate that changes in the orexin system are associated with depression and that these changes may be sexually dimorphic. However, in contrast to aforementioned studies, one study of 15 patients reported that there were no differences in CSF orexin levels between patients with major depressive disorder (MDD), those with manic disorder, and control participants [39]. Although these data are not consistent with those from other studies, the number of participants was low and therefore may have been insufficient to detect statistical differences in orexin levels between groups.

Taken collectively, evidence suggests that changes in orexin levels or orexin signalling may be associated with depression; however, given the heterogeneity of the findings by animal and human studies, further research is needed to determine the role of orexins and their receptors in depression.

#### 3.4.2. Anxiety

Orexins play a role in the modulation of stress responses and have been implicated in promoting anxiety-like behaviours in rats, with both OX1R and OX2R playing a role [17,40]. Administration of orexin has been shown to induce panic-like behaviours in several animal models [41,42]. Conversely, treatment with OXR1 antagonists reduced panic-like behaviour and cardiovascular responses to anxiety-inducing stimuli in an animal model of panic [43,44,45,46]. Additionally, stimulation of OX2Rs induced anxiolytic behaviours, demonstrated by reduced fear conditioning and conflict freezing and startle in a rat model of anxiety [47].

Induction of anxiety-like behaviours is mediated by projections of orexin neurons to regions involved in mediating stress responses, including the amygdala and bed nucleus of stria terminalis [48,49]. Additionally, anxiety-promoting role of orexin has been shown to be mediated by projections to the paraventricular thalamus and LC [41].

An increase in orexin-A in anxiety has also been demonstrated in humans. In a study of 56 adolescents diagnosed with an anxiety disorder and 32 healthy controls, orexin-A levels were significantly higher in those with anxiety [50]. Furthermore, there was a positive correlation trend between trait anxiety and orexin-A [50]. Another study in human subjects reported that variations in HCRTR1 gene, which mediates hyperarousal, was associated with the aetiology of panic disorder and agoraphobia. Specifically, the presence of HCRTR1 rs2271933 allele was significantly associated with panic disorder/agoraphobia, with the association being particularly strong in female participants [51].

#### 3.4.3. Addiction

The role of orexins in reward-seeking behaviours is closely related to its role in addiction, which has been extensively summarized in other reviews and will only be briefly summarized here (for example, see [3,4,7]. Studies of the orexin system reported that orexin neurons in the lateral hypothalamus are activated by drugs of abuse (i.e., cocaine, morphine, and fentanyl) [52,53,54]. Further, evidence shows that seeking for all major addictive drugs is more involved than by natural non-drug reinforcers [3]. Studies have shown that prepro-orexin mRNA is upregulated following alcohol consumption, or heroin self-administration. Further, acute withdrawals from morphine and heroin, as well as intermittent access to fentanyl also increase orexin expression. Conversely, administration of OXR1 antagonist SB334867 reduces intake of oxycodone, heroin, fentanyl, and remifentanil, and reversed addiction behaviours induced by intermittent access to fentanyl [3,53].

Postmortem analysis of human brains has shown a 54% increase in the number of orexin-producing neurons in brains of heroin addicts compared with controls, with an average orexin-producing cell being 22% smaller in addicts compared with controls [55]. Taken together with studies in animals, these findings indicate that addiction may be associated with dysregulation of orexin production and/or signalling.

#### 3.4.4. ADHD

Studies evaluating the role of orexin in ADHD are limited, however, it has been demonstrated that orexin A levels were significantly lower in children with ADHD than those without ADHD [56]. Further, a study of patients with narcolepsy showed that those with orexin deficiency (i.e., narcolepsy type 1) displayed higher severity of ADHD hyperactive domain and depressive symptoms compared with those without orexin deficiency (i.e., narcolepsy type 2) [57].

#### 3.4.5. Schizophrenia

Although the evidence for the potential role of the orexin system in schizophrenia is limited, some studies indicate that the orexin system may contribute to the neurochemical alterations associated with schizophrenia. In an animal model of schizophrenia, characterized by an increased dopamine neuron activity, systemic administration of dual orexin receptor normalized the increase in dopamine neuron activity [58]. Similar findings were observed following the administration of the orexin receptor antagonist into the paraventricular nucleus of the thalamus, a region that provides peptidergic inputs to the NAc. Additionally, studies in humans have shown that orexin levels may be altered in individuals with schizophrenia. One study has shown that individuals with schizophrenia treated with antipsychotic drug haloperidol have lower CSF orexin A levels compared to those who were not medicated [59].

Another study demonstrated that patients with schizophrenia had significantly higher plasma orexin A levels than healthy individuals [60]. Further, the same study showed that patients with high orexin A levels had significantly fewer negative and disorganized symptoms and tended to have fewer perseverative errors than those with normal orexin A levels.

#### 3.4.6. Overview of Hedonic Tone and Anhedonia

As reviewed previously, anhedonia is the state of reduced ability to express feelings of pleasure [61]. Anhedonia is characterized by deficits in reward-related processing, which may present as loss of interest or pleasure and may impede an individual’s ability to engage in goal-directed behaviours. Pathological hedonic dysfunction may contribute to affect-related disorders, including depression and addiction [62]. Hedonic tone, also referred to as hedonic capacity or hedonic responsiveness, is the trait or genetic predisposition that underlies an individual’s baseline range and lifelong characteristic ability to feel pleasure. Low hedonic tone represents a reduced capacity to experience pleasure, thus increasing the likelihood of experiencing anhedonia. It has been proposed that individuals with low hedonic tone may have an increased need for stimulation, which may manifest as seeking external stimulation (e.g., risky behaviour or substance abuse) or internal stimulation (e.g., fantasy) to raise their hedonic tone [61]. Individuals with low hedonic tone may attempt to cope by maximizing pleasure or raise mood from their low baseline tone. In the absence of such stimulation, individuals may experience a shift toward their usual, lower hedonic tone, and therefore a drop in their mood to their baseline dysphoric state.

The interplay between hedonic tone and the experience of reward indicates that hedonic tone may be regulated by neural pathways involved in processing reward. Accordingly, studies of neurobiology of hedonic tone have identified dopaminergic circuits as playing a key role in the maintenance of hedonic tone [61]. Specifically, hedonic tone is maintained by circuits that contain bottom-up and top-down projections into the prefrontal cortex, lateral habenula, and the VTA dopamine system. Regions presumed to contribute to modulation of hedonic tone are summarized in Figure 1. Hedonic tone is closely related to mood, reward, and motivation, and is modulated by the limbic-cortical-striatal-pallidal-thalamic (LCSPT) circuits. In healthy individuals, presentation of positive stimuli increases the activation of regions involved in reward processing, including the caudate, putamen, NA, basal forebrain, medial frontal region, anterior cingulate cortex, inferior parietal area, right fusiform, and lingual gyrus [63,64,65]. In contrast, suppressing a positive emotion is associated with activation of the right ventrolateral prefrontal cortex [66]. Furthermore, anhedonia is negatively correlated with the activation of the NAc, basal forebrain, and the hypothalamus, and positively correlated with the activity in the ventromedial prefrontal cortex [64,67,68]. These findings indicate that anhedonia alters the activity of regions involved in reward processing and may be partly due to the insufficient activation of circuits that regulate feelings of pleasure.

Low hedonic tone has been associated with several psychiatric disorders, including MDD, substance use, and schizophrenia [61]. For example, evidence indicates that neural circuits involved in maintaining hedonic tone may be altered in individuals with depression. Patients with MDD show attenuated activation of the ventral striatum, medial frontal cortex, and the NA, and increased activation in the inferior frontal cortex, subgenual anterior cingulate cortex, thalamus, putamen, and the insula [61]. These findings suggest that alteration of neural pathways involved in the maintenance of hedonic tone may underlie changes in hedonic tone observed in patients with psychiatric disorders.

## 4. Potential Role of Orexin in Modulation of Hedonic Tone and Anhedonia

Although the role of orexin in the modulation of hedonic tone has not been specifically investigated, studies in animal models indicate that orexin is involved in the modulation of hedonic behaviour, as it relates to food intake and drug abuse. Additionally, a mounting body of evidence indicates that mood disorders characterized by low hedonic tone, such as depression, anxiety, addiction, and ADHD, are also associated with alterations in orexin signalling [3,4,17,32,33,34,40,47,56]. Further, projections of orexin-containing neurons overlap circuits involved in the regulation of hedonic tone and orexin administration in these areas have been shown to affect mood and hedonic behaviours.

Evidence for orexin in motivated and reward-seeking behaviours, as well as evidence showing alterations in orexin signalling in mood disorders that are characterized by low hedonic tone, suggests that the orexin system may also be involved in the maintenance of hedonic tone. More specifically, orexin may potentiate hedonic behaviours (e.g., intake of food or drugs of abuse) by increasing the feeling of pleasure or reward to various stimuli [4]. Conversely, dysregulation of orexin signalling may underlie low hedonic tone or anhedonia, such as that perceived in orexin may potentiate hedonic behaviours by increasing the feeling of pleasure or reward to various stimuli. Conversely, dysregulation of orexin signalling may underlie low hedonic tone or anhedonia, such as that perceived in MDD, anxiety, or ADHD.

Neural circuits involved in the regulation of hedonic tone overlap with the projections of orexin-containing neurons (Figure 1). Further, orexin has been shown to have direct effects on neurons in several of these regions. Studies showing direct actions of orexin neurons in these regions in animal models are summarized in Table 1. A large body of evidence has shown that direct administration of orexin into the NAc, VTA, medial prefrontal cortex, and the ventral pallidum affects neuronal activity in animal models and elicits a wide variety of behaviours, including drug intake, fear, conditioned place preference, and motivation for drug administration [69,70,71,72]. Evidence shows that orexin A increases the activity of dopaminergic neurons in the VTA and dopamine release in the NAc, indicating that VTA-NAc dopamine circuits may mediate the effects of orexin on hedonic behaviours [70,71,73,74,75,76,77]. Further, orexin administration into the ventral pallidum and mPFC has also been shown to elicit hedonic behaviours, including sucrose and drug (i.e., opioid and alcohol) [69] intake [72,78]. Notably, a study that used a rat model of maternal separation showed that early life stress increases the expression of OXR1 and OXR2 levels in the PFC.

Additional evidence for orexin mediating hedonic behaviours comes from a study in rats that evaluated the effects of orexin-A or opioid stimulation of cortical sites on “liking” reactions and compared the ability of opioid and orexin stimulation to alter the motivation to consume a sweet food [62]. The study identified two “hedonic hotspots”, one in the rostral orbitofrontal cortex (OFC), and another one in the insula, where microinjection of orexin-A increased the number of positive hedonic reactions elicited by sucrose. The effect of orexin was specific to the positive, hedonic behaviour, as orexin microinjections had no effect on negative “disgust” reactions elicited by bitter taste. Conversely, microinjections of orexin in the caudal OFC suppressed the “liking behaviours in the same rats. Further, microinjection of orexin-A into the rostral OFC hedonic hotspot activated neurons in the NAc shell, and microinjection of orexin A into the insula activated neurons in the ventral pallidum and NAc shell. These findings indicate that orexin may mediate hedonic behaviour via actions on the OFC, which, in turn, activates the NAc and the ventral pallidum.

Taken collectively, evidence indicates a strong overlap between circuits that regulate hedonic tone and orexin-containing neurons and suggests that orexin signalling may be involved in the modulation of hedonic tone and may contribute to disorders characterized by low hedonic tone. Further research must look into the potential role of Orexin related agents in the treatment resistant depression. Additionally, studies are needed to elucidate other potential targets of the orexin system that may modulate hedonic tone. One potential target of the orexin system is the rostromedial tegmentum, a region that provides a strong inhibitory input to dopaminergic neurons in the VTA and plays a critical role in behavioural inhibition [104].

## 5. Summary and Conclusions

In summary, orexin peptides are involved in the regulation of a wide variety of physiological functions and behaviours, ranging from autonomic functions such as food intake and cardiovascular regulation to modulation of higher cognitive functions and mood. Although the role of orexin in modulating hedonic tone has not been specifically investigated, a large body of evidence indicates that orexin may potentiate hedonic behaviours by increasing the feeling of pleasure or reward to various stimuli. Conversely, dysregulation of orexin signalling may underlie low hedonic tone or anhedonia, such as that observed in depression, anxiety, ADHD, and addiction. Further, orexin appears to play a key role in regulating behaviours in motivationally charged situations, such as food-seeking during hunger, or drug-seeking during withdrawal. Therefore, it would be expected that dysregulation of orexin expression or signalling is associated with changes in hedonic tone. Further studies investigating this association are warranted.

## Figures and Tables

**Figure 1 brainsci-12-00150-f001:**
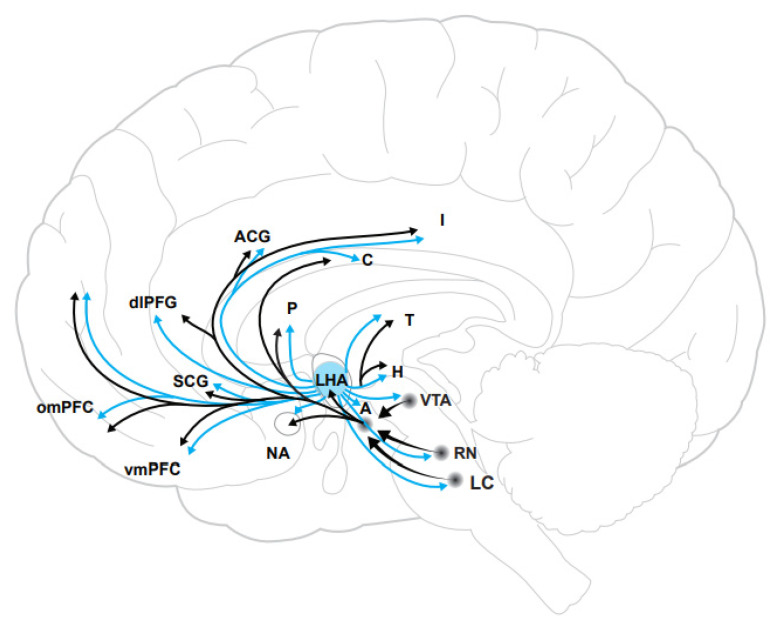
A schematic diagram showing the overlap of orexin projections with hypothesized regions contributing to modulation of hedonic tone. Orexin neurons are shown in blue. Abbreviations: A, amygdala; ACG, anterior cingulate cortex; C, caudate; DLPFG, dorsolateral prefrontal cortex; H, habenula; I, insula; LC, locus coeruleus; LHA = lateral hypothalamus; NA, nucleus accumbens; OMPFC, orbitomedial prefrontal cortex; P, putamen; RN, raphe nucleus; SCG, subgenual cingulate cortex; T, thalamus; VTA, ventral tegmental area; and VMPFC, ventromedial prefrontal cortex.

**Table 1 brainsci-12-00150-t001:** Summary of animal studies showing behavioural effects of orexins in regions involved in the regulation of hedonic tone.

Reference	Effects of Orexin
** *Nucleus Accumbens* **	
Assar et al., 2019 [79]	Administration of OX1R antagonists reduced acquisition of morphine sensitization OX2R antagonist produced a similar effect, but at a higher dose
Fartootzadeh, 2019 [80]	Administration of OX2R antagonists in NAcc inhibited nicotine-induced increase in neuronal excitation
Lai et al., 2018 [74]	Activation of OX1R by orexin A facilitated sucrose-stimulated DA transmission by increasing the basal activity of VTA DA neurons
Lei et al., 2016 [69]	Administration of OX1R antagonist in the medial NAcc shell and mPFC significantly reduced excessive alcohol intake
Castro et al., 2016 [70]	Orexin enhances sucrose “liking” and intake but scopolamine in the caudal shell shifts “liking” toward “disgust” and “fear”
Liu et al., 2020 [71]	Microinjection orexin-A significantly increased palatable food intake; the effect was inhibited by pretreatment with OX1R antagonist
Mayannavar et al., 2016 [81]	Microinjection of orexin A antagonist in the NAc reduced water and alcohol intake, but did not affect preference to alcohol
Patyal et al., 2012 [73]	Application of Orexin A increased dopamine release in the NAcc shell without altering reuptake at dopamine terminals, indicating that locally released orexin A can modulate dopamine release in NAcc shell
Sahafzadeh et al., 2018 [82]	Administration of OX1R and OX2R antagonists into the NAcc attenuated the effect of food deprivation on morphine reinstatement
Kwok, C et al., 2021 [83]	OX1R inhibition in NAcc shell altered alcohol intake in male, but not female mice. OX1R inhibition reduced compulsion-like alcohol intake in both sexes
Lei, K., et al., 2019 [84]	Activation of OX1Rs promoted alcohol intake during intermittent-access
** *Ventral Tegmental Area* **	
Azizbeigi et al., 2019 [85]	OX1R antagonist suppressed morphine reinstatement induced by stress or drug priming
Azizbeigi et al., 2019 [86]	OX2R antagonist suppressed morphine reinstatement induced by stress or drug priming
Taslimi et al., 2012 [87]	Orexin A elicited conditioned place preference; the response was inhibited by administration of D1 and D2 receptor antagonists into the NAcc
Yazdi et al., 2015 [88]	Administration of OX2R antagonist into the VTA or NAcc dose-dependently inhibited the development of LH stimulation-induced conditioned place preference Co-administration of low doses of OX2R antagonist and CB1 receptor antagonist into the NAcc reduced conditioning scores
Azizi et al., 2018 [89]	OX2R antagonist inhibited the development of nicotine-induced conditioned place preference
Bernstein et al., 2018 [90]	OX1R knock-down delayed cocaine self-administration, indicating that OX1R are involved in motivation for cocaine
Brown et al., 2016 [91]	OX1R antagonist attenuated cue-induced reinstatement of ethanol-seeking
España et al., 2011 [75]	Orexin 1 increased the effect of cocaine on tonic and phasic DA signaling and increased the motivation to self-administer cocaine
Farahimanesh et al., 2017 [92]	OXR1 and OXR2 antagonists attenuated morphine conditioned place preference acquisition during the conditioning phase, and expression during the post-conditioning phase
James et al., 2011 [93]	OXR1 antagonist dose-dependently attenuated cue-induced reinstatement of cocaine-seeking
Moorman et al., 2010 [76]	Orexin (not specified if A or B) increased the activity of DA neurons and augmented excitatory responses to mPFC stimulation OX1R antagonist decreased tonic DA cell activity during active but not rest period
Muschamp et al., 2014 [94]	OX1R antagonism increased the threshold for intracranial self-stimulation; the response was blocked by a dynorphin receptor antagonist OXR1 antagonism reduced cocaine-induced impulsive behaviour Orexin A excited DA neurons in the VTA OX1R antagonist in the VTA reduced cocaine intake
Naghavi et al., 2019 [77]	Administration of D1 and D2 receptor antagonists attenuated the acquisition of place preference by orexin A
Olney et al., 2017 [95]	OX1R antagonist reduced binge-like ethanol intake but did not affect sucrose intake
Richardson et al., 2012 [96]	OX1R antagonist administration attenuated the morphine conditioned place preference score induced by administration of carbachol into the LGA, indicating that OX1R plays a role in sensitization to morphine
Saferi et al., 2019 [97]	OX1R and OXR2 antagonists reduced antinociceptive effect induced by carbachol administration into the lateral hypothalamus
Srinivasan et al., 2012 [98]	Administration of a dual OX1R and OXR2 antagonist into the VTA decreased operant self-administration of ethanol but not and sucrose Orexin A increased firing of VTA neurons
Taslimi et al., 2012 [87]	Orexin A administration induced conditioned place preference; the effect was inhibited by administration of D1 and D2 receptor antagonists
Terrill et al., 2016 [99]	Orexin-A increased intake of palatable food (high-fat food and sucrose), whereas OX1R antagonist suppressed sucrose intake
Wang et al., 2009 [100]	Orexin-A administration reinstated cocaine seeking and caused release of glutamate and dopamine in the VTA
Zarepour et al., 2014 [101]	OX1R antagonist inhibited the acquisition but not expression of LH stimulation-induced morphine conditioned place preference
** *Ventral Pallidum* **	
Ho et al., 2013 [72]	Orexin amplified hedonic liking for sweetness
Mohammadkhani et al., [78]	OXR1 antagonist decreased motivation for remifentanil without affecting remifentanil consumption Thus, Orexin amplifies motivation to gain reward from drugs and sucrose.
** *Medial prefrontal cortex* **	
Lei et al., 2016 [69]	OX1R antagonist reduced alcohol drinking
Cole et al., 2015 [102]	Systemic OX1R antagonist significantly reduced cue-driven consumption in sated rats and increased Fos expression in mPFC
Dimatelis et al., 2018 [103]	Maternal separation increased OXR1 and OXR2 levels in the PFC
Lambe et al., 2005 [29]	Similar to nicotine, orexin B infusion into the PFC improved accuracy under high attentional demand

## Data Availability

Not Applicable.

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
