# Peer review of "Neurobiology of the Orexin System and Its Potential Role in the Regulation of Hedonic Tone"

_brainsci, 2022, doi:10.3390/brainsci12020150_

Round 1

Reviewer 1 Report

The authors provided a nice review on the orexin system and its potential role in the regulation of hedonic tone. Overall, the manuscript is well written and interesting. Since a major portion of the manuscript is about VTA DA neurons, it would be nice to add a little discussion on how the RMTg, one of the major inhibitory inputs to VTA DA neurons, might be receiving orexin inputs and plays a role in the process.

Reviewer 2 Report

The review describes in details orexin system and highlights a potential connection between orexin system and hedonic behaviour. The manuscript is divided into several parts describing separately orexin system, hedonic tone and anhedonia and finally explains potential role of orexin system in modulation of hedonic behaviour.

Major issues:

  • Part called “Influence of orexin on mood and mood disorders” – authors do not describe only mood disorders but also other types of mental health condition. Therefore, I would recommend authors to change the title of this part.
  • Since authors mentioned different mental disorders and their connection to orexin system, I would propose to also add schizophrenia and role of orexin system in such severe mental illness.
  • Part called “Depression” – I would highly recommend authors to use terms depressive-like symptoms or particular symptoms observed in animals instead of “depressive symptoms”. Authors may use depressive phenotype too. In case of term “animals with depression”, authors should use “animal model of depression”.

Minor issues:

  • Line 13: word “rpocue”?
  • Line 188, 283, 285 and 309: acronym MDD is missing
  • Line 314: VTA acronym is missing
